# Depth-Width Trade-offs for ReLU Networks via Sharkovsky's Theorem

**Vaggos Chatziafratis**
Department of Computer Science
Stanford University

**Sai Ganesh Nagarajan & Ioannis Panageas & Xiao Wang**
Singapore University of Technology and Design

## Abstract

Understanding the representational power of Deep Neural Networks (DNNs) and how their structural properties (e.g., depth, width, type of activation unit) affect the functions they can compute, has been an important yet challenging question in deep learning and approximation theory. In a seminal paper, Telgarsky highlighted the benefits of depth by presenting a family of functions (based on simple triangular waves) for which DNNs achieve zero classification error, whereas shallow networks with fewer than exponentially many nodes incur constant error. Even though Telgarsky's work reveals the limitations of shallow neural networks, it doesn't inform us on *why* these functions are difficult to represent and in fact he states it as a tantalizing open question to characterize those functions that cannot be well-approximated by smaller depths.

In this work, we point to a new connection between DNNs expressivity and Sharkovsky's Theorem from dynamical systems, that enables us to characterize the depth-width trade-offs of ReLU networks for representing functions based on the presence of a generalized notion of *fixed* points, called *periodic* points (a fixed point is a point of period 1). Motivated by our observation that the triangle waves used in Telgarsky's work contain points of period 3 – a period that is special in that it implies chaotic behaviour based on the celebrated result by Li-Yorke – we proceed to give general lower bounds for the width needed to represent periodic functions as a function of the depth. Technically, the crux of our approach is based on an eigenvalue analysis of the dynamical systems associated with such functions.

## 1 Introduction

In approximation theory, one typically tries to understand how to best approximate a complicated family of functions using simpler functions as building blocks. For instance, Weierstrass (1885) proved a general result stating that every continuous function can be uniformly approximated as closely as desired by a polynomial. It wasn't until later that Vitushkin (1959) gave quantitative bounds between the approximation error and the polynomial's degree. Drifting away from polynomials and given the recent breakthroughs of deep learning in a variety of difficult tasks like image classification, natural language processing, game playing and self-driving cars, researchers have tried to understand the approximation theory that governs neural networks. This question of *neural network expressivity*, i.e. how architectural properties like the depth, width or the activation units affect the functions it can compute, has been a fundamental ongoing challenge with a rich history. A classical result by Cybenko (1989), Hornik et al. (1989), Fukushima (1980) demonstrates the expressive power of neural networks: it states that even two layered neural networks (using well known activation functions) can approximate any continuous function on a bounded domain. The caveat is that the size of such networks may be exponential in the dimension of the input, which makes them highly susceptible to overfitting as well as impractical, since one can always add extra layers in their model aiming at increasing the representational power of the neural network.

More recently, in a seminal paper by Telgarsky (2016), it was shown that there exist functions that can be represented by DNNs, i.e, by some particular choice of weights on their edges (and for a wide variety of standard activation units in their layers), yet cannot be approximated by shallow networks unless they are exponentially large. More concretely, he showed that for any positive integer $k$, there exist neural networks with $\Theta(k^3)$ layers, $\Theta(1)$ nodes per layer, and $\Theta(1)$ distinct parameters which cannot be approximated by networks with $\mathcal{O}(k)$ layers, unless they have $\Omega(2^k)$ nodes. At a high level, he uses the number of oscillations present in certain functions as a notion of "complexity" that distinguishes between deep and shallow networks' representation capabilities via the following three facts: a) functions with few oscillations poorly approximate functions with many oscillations, b) functions computed by networks with few layers must have few oscillations and c) functions computed by networks with many layers can have many oscillations.

Our main contribution is a novel connection between the theory of dynamical systems and the representational power of DNNs via the well-studied notion of *periodic points*, a notion that captures the important notion of *fixed points* of a continuous function.

**Definition 1** (Period). *We say that a (continuous) Lipschitz function $f : [0, 1] \to [0, 1]$ contains a point of period $n \geq 1$ if there exists a point $x_0 \in [0, 1]$ such that[1]:*

$$f^n(x_0) = x_0 \quad and \qquad\qquad\qquad \text{(point of period } n\text{)}$$
$$f^k(x_0) \neq x_0, \ \ \forall \ 1 \leq k \leq n - 1.$$

*In particular, all numbers in $C = \{x_0, f(x_0), f(f(x_0)), \ldots, f^{n-1}(x_0)\}$ are distinct, each of which is a point of period $n$ and the set $C$ is called a cycle (or orbit) of period $n$. Observe that since $f : [0, 1] \to [0, 1]$ is continuous, it certainly has at least one point of period 1, which is called a fixed point.*

For the rest of this paper, we focus on (continuous) Lipschitz functions $f : [0, 1] \to [0, 1]$, unless otherwise stated. Note that the choice of interval $[0, 1]$ is for simplicity of our presentation and that our results will hold for any closed interval $[a, b]$.

As we observe, points of period 3 are contained in both Telgarsky (2016) and Schmitt (2000) constructions and this could as well have been a coincidence, however we show that *the existence of periodic points of certain periods are actually one of the reasons explaining why depth is needed to represent functions that contain them (otherwise exponential width is required)*. Towards this direction, we will make use of a deep result in the literature of iterated dynamical systems, that gave rise to the field of Combinatorial Dynamics (Alseda et al. (2000)), called Sharkovsky's Theorem (Sharkovsky (1964; 1965)).

## 1.1 SHARKOVSKY'S THEOREM

Consider the set of positive natural numbers $\mathbb{N}^* = \{1, 2, \ldots\}$ and define the following (decreasing) ordering $\triangleright$ called *Sharkovsky's ordering* as follows:

$$3 \triangleright 5 \triangleright 7 \triangleright \cdots \triangleright \text{ (odd numbers bigger than one)}$$
$$\triangleright 2 \cdot 3 \triangleright 2 \cdot 5 \triangleright 2 \cdot 7 \triangleright \cdots \triangleright \text{ (odd multiples of two but not two)}$$
$$\triangleright 2^2 \cdot 3 \triangleright 2^2 \cdot 5 \triangleright 2^2 \cdot 7 \triangleright \cdots \triangleright \text{ (odd multiples of four but not four)}$$
$$\vdots$$
$$\triangleright \cdots \triangleright 2^4 \triangleright 2^3 \triangleright 2^2 \triangleright 2 \triangleright 1 \text{ (powers of two in decreasing order).}$$

This is a total ordering; we write $l \triangleright r$ or $r \triangleleft l$ whenever $l$ is to the left of $r$. Sharkovsky showed that this ordering describes which numbers can be periods for a continuous map on an interval; allowed periods need to be a suffix of the Sharkovsky ordering:

**Theorem 1.1** (Sharkovsky's "Forcing" Theorem, Sharkovsky (1964; 1965)). *Let $I$ be a closed interval and $f : I \to I$ be a continuous map. If $n$ is a period for $f$ and $n \triangleright n'$, then $n'$ is also a period for $f$.*

*Remark* 1.1. Note that the number 3 is the maximum period according to Sharkovsky's ordering, so an important corollary is that a function having a point of period 3, must also have points of any

---

[1]As usual, $f^n(x_0)$ denotes the composition of $f$ with itself $n$ times, evaluated at point $x_0$.

period. This special corollary is a weaker version of Sharkovsky's theorem and was proved some years later[2] in a celebrated result by Li & Yorke (1975), who coined the term "chaos" as used in Mathematics.

We conclude the subsection with the definition of a *prime period* of a function $f$.

**Definition 2** (Prime period). *A function $f$ has prime period $n$ as long as it has a cycle of period $n$, but has no cycles with period greater than $n$ according to the Sharkovsky ordering.*

For example, in the interval [0,1], the function $f(x) = 1 - x$ has prime period 2, since $f(f(x)) = 1 - (1 - x) = x$ so all points are periodic with period 2, except the fixed point at 1/2.

Before formally stating our main theorems, we present an illustrative example inspired from Telgarsky's triangle wave construction and we connect it to DNNs' sensitivity to weight perturbations and their representational power.

## 1.2 SENSITIVITY ANALYSIS - A MOTIVATING EXAMPLE

An important ingredient in Telgarsky's proof, was the "triangular wave" function (sometimes referred to as the tent map or sawtooth) depicted in Figure 1b and given by:

$$t(x; 2) = \begin{cases} 2x, & \text{if } 0 \leq x \leq \frac{1}{2} \\ 2(1 - x), & \text{if } \frac{1}{2} < x \leq 1 \\ 0, & \text{otherwise} \end{cases}$$

He shows that the composition of $t(x; 2)$ with itself $k$ times (denoted by $t^k(x; 2)$), will create exponentially (in $k$) many oscillations and as a result he is able to show a separation for the classification error when using a shallow vs a deep neural network as a predictor.

Our starting point is the observation that the triangular wave function $t(x; 2)$ contains points of period 3, e.g. $(\frac{2}{9} \to \frac{4}{9} \to \frac{8}{9} \to \frac{2}{9})$. It follows in particular, that $t(x; 2)$ exhibits Li-Yorke Chaos (Li & Yorke (1975)) in the sense that it contains all periods. The compositions of such functions will look highly complex (see Figure 2) and in fact Telgarsky heavily relied on the highly oscillatory behavior of $t(x; 2)$ to prove his depth separation result.

However, his result doesn't inform us on what would happen if one used a slightly modified version of the triangle wave $t(x; 2)$. Observe that since a simple neural network with one hidden layer can represent the function $t(x; 2)$, the question is basically equivalent to asking how modifying the weights on the edges of the neural network can affect its representational power (see Figure 1), hence the title of the current subsection. The main question is can we have a general theory that informs us on when will the function composition be hard to represent and when not? Our paper's main point is to provide an answer by checking if the function at hand has a simple property, relating to the presence of chaotic behavior.

To illustrate our point, consider the generalized triangle wave function $t(x; \mu)$ parameterized by $\mu$:

$$t(x; \mu) = \begin{cases} \mu x, & \text{if } 0 \leq x \leq \frac{1}{2} \\ \mu(1 - x), & \text{if } \frac{1}{2} < x \leq 1 \\ 0, & \text{otherwise} \end{cases}$$

This function parameterized by $\mu$ ranges from $[0, \mu/2]$ and is closely related to the logistic map $f(x) := rx(1 - x)$ used in Schmitt (2000) and exhibits a variety of limiting behaviors: for instance, it converges to a stable fixed point when $\mu \leq 1$, it exhibits chaos when $\mu = 2$ etc.[3] Instead of $\mu = 2$, if we set $\mu = 1$, we get the network depicted in Figure 1c, 1d.

---

[2]Due to historical reasons during the late 20th century, the theory of dynamical systems saw a parallel development in the USA and the USSR, hence Sharkovsky's theorem (1964) remained unknown in the USA, until in 1975 a weaker version was rediscovered by James Yorke and his graduate student Tien-Yien Li, in their celebrated paper called "Period Three Implies Chaos".

[3]For more, the interested reader can also check `https://en.wikipedia.org/wiki/Logistic_map`.

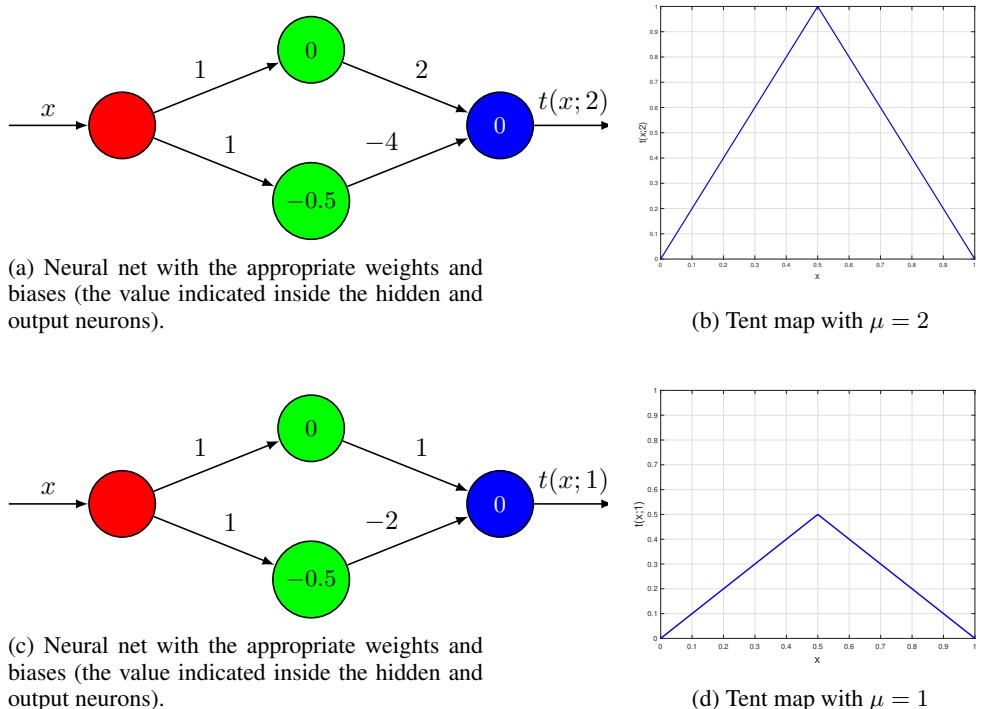

(a) Neural net with the appropriate weights and biases (the value indicated inside the hidden and output neurons).

(b) Tent map with $\mu = 2$

(c) Neural net with the appropriate weights and biases (the value indicated inside the hidden and output neurons).

(d) Tent map with $\mu = 1$

Figure 1: The neural network instantiations that are used to create two different tent maps which vary only in the maximum value. This is effected by a small change of weights in the output layer. All activation functions are ReLU's.

Note that compositions of $t(x; 1)$ (created by the same neural network architecture but with slightly different weights), behave completely differently since in the $\mu = 1$ case, we will not get a highly oscillatory behavior. This can be seen in Figure 3. One difference between the two cases is the relative position of the map with the line $y = x$ and this seems to be pointing that fixed points and their generalizations i.e. periodic orbits play an important role when dealing with function compositions. Indeed, despite the wide range of possibilities one can expect by composing such functions, as we show, their behavior can be characterized using tools from dynamical systems; the exponential growth in complexity (or lack thereof) of these compositions can be explained by invoking a fundamental property of these continuous functions on bounded intervals which is the existence (or not) of periodic points of certain periods.

Similarly, we can argue about changing the parameters of the logistic map which is given by $f(x; r) := rx(1 - x)$ used in Schmitt (2000) for sigmoidal networks (where $f(x; 4)$ was used). The properties of the logistic map are well known and was first studied by Robert May and Mitchell Feigenbaum (May (1976) and Feigenbaum (1976)). It is known that as one varies the parameter $r$, the logistic map gives rise to a plethora of different behaviors, hence the same is true for when one slightly perturbs the weights of a neural net used to represent the map. Please refer to Appendix C for some figures that illuminate these differences in the logistic map.

### 1.3 Informal Statements of Main Theorems

We demonstrate that a simple property of $f$ governs the depth-width trade-offs in order to represent it and we give quantitative bounds for them. This simple property has to do with the periods that the function $f$ contains. Informally, our first main theorem states that if a function $f$ contains periodic points with certain periods, then composing $f$ with itself many times, will result in exponentially many oscillations, giving rise to complicated behaviors and chaos:

**Theorem 1.2.** *Let $f : [0, 1] \to [0, 1]$ be a continuous function. Assume that there exists a cycle of period $n$, where $n = m \cdot p$ with $p$ being an odd number greater than one and with $m$ being a*

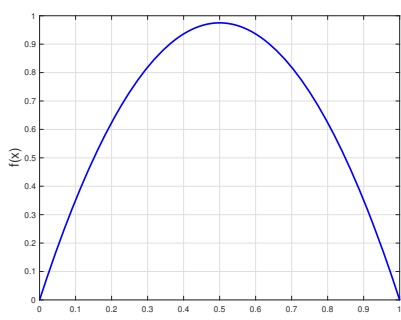

(a) $f(x) := 3.9 * x(1-x)$ on the interval $[0,1]$

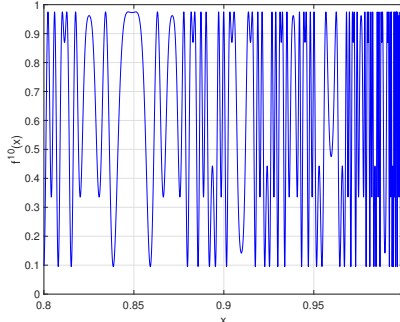

(b) $f^{10}(x)$ of the map on the left (zoomed in)

Figure 2: Compositions of the logistic map $f(x) = 3.9x(1-x)$ defined on the interval $[0,1]$. This map is well known to exhibit chaos and in the above figure has non-vanishing oscillations that grow with the number of compositions, albeit irregularly.

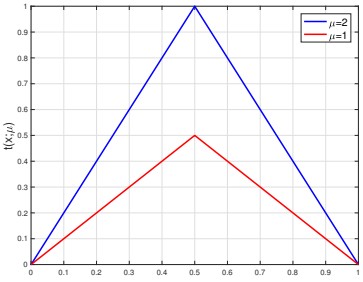

(a) $t(x;\mu)$ for the tent map with $\mu = 2$ (blue) and $\mu = 1$ (red)

(b) $t^6(x;\mu)$ for the tent map with $\mu = 2$ (blue) and $\mu = 1$ (red).

Figure 3: Compositions of $t(x;\mu)$ with different parameters $\mu = 2$ and $\mu = 1$ are shown. The compositions create (exponential) non-vanishing oscillations when $\mu = 2$, however the compositions remain unchanged when $\mu = 1$.

*power of two (it might be $m = 1$). Then, there exist $x, y \in [0, 1]$ such that the function $f^{mt}$ (taking $mt$ compositions of $f$ with itself) "oscillates" (also look Definition 4) at least $\rho^t$ times between $x$ and $y$ for all $t \in \mathbb{N}^*$, where $\rho$ is the positive root greater than one of the polynomial equation $\lambda^{p-1} - \lambda^{p-2} - 1 = 0$.*

Our second main theorem then draws the connection between the number of oscillations a function has and the depth-width trade-offs needed:

**Theorem 1.3.** *Let $k$ be a positive integer and $f$ be a function as above. We set $\rho$ to be the positive root greater than one of the polynomial equation $\lambda^{p-1} - \lambda^{p-2} - 1 = 0$. We can construct a sequence of points $(x_i, y_i)_{i=1}^{2n}$ with $n := \frac{\lfloor \rho^k \rfloor}{2}$ such that the classification error of the function $f^{mk}$ is zero, whereas the classification error of any neural network with $l$ layers and $u$ nodes per layer, where $u \leq \frac{\rho^{\frac{k}{l}}}{8}$, necessarily has classification error $\geq \frac{1}{4}$.*

Formal statements for the two theorems can be found in Section 3 and Section 4.

Using these theorems, we draw connections with previous results Telgarsky (2016), Schmitt (2000) in a unified way, thus identifying chaotic behavior as the main underlying thread for depth-width trade-offs. Technically, our approach is based on an eigenvalue analysis of certain matrices associated with such periodic functions.

## 1.4   OTHER RELATED WORK

Understanding the benefits of depths on the expressive power a specific computational model can have, is an important area of research spanning different computational models and results come in the flavor of depth separation arguments. Roughly speaking, many of the results in this area rely on a suitably defined notion of "complexity" of a function we would like to represent, and then proceed by proving that under this notion, deep models have significantly more power than shallower models. For example, if the computational model of interest is the family of boolean or threshold circuits, depth lower bounds are given in Hastad (1986); Rossman et al. (2015); Håstad (1987); Parberry et al. (1994); Kane & Williams (2016). Furthermore, people have analyzed sum-product networks (summation and product nodes) and studied trade-offs for depth (Delalleau & Bengio (2011); Martens & Medabalimi (2014)).

Coming closer to neural networks computation where the activation units can be general real-valued functions, important previous results include Eldan & Shamir (2016); Telgarsky (2015; 2016); Schmitt (2000); Montufar et al. (2014); Malach & Shalev-Shwartz (2019); Poole et al. (2016); Raghu et al. (2017); Arora et al. (2016); Liang & Srikant (2016); Kileel et al. (2019). Regarding the aforementioned notions of "complexity" used in depth separation arguments, examples include the notion of global curvature (Poole et al. (2016)), trajectory length (Raghu et al. (2017)), number of oscillations (Telgarsky (2015; 2016) and Schmitt (2000)), number of linear regions (Montufar et al. (2014)), fractals (Malach & Shalev-Shwartz (2019)), dimensions for algebraic varieties (Kileel et al. (2019)) and more. Our work is more closely related to Telgarsky (2015; 2016), and Schmitt (2000) since it is easy to see that their maps are chaotic, but we conjecture that many of the notions of complexity introduced in this line of research to showcase benefits of depth actually arise due to chaotic behavior. In this sense, we conjecture that chaotic behavior is the main culprit for the failure of neural networks to represent certain functions, unless they are sufficiently deep (or have exponential width). Moreover, other works that have exploited the powerful result by Li-Yorke (in online learning frameworks) are Palaiopanos et al. (2017); Chotibut et al. (2019).

## 2   FURTHER BACKGROUND: THE COVERING LEMMA

The crux of the proof of Sharkovsky's theorem provided by Burns & Hasselblatt (2011) contains a covering lemma that will be our starting point to prove our main results. Before we proceed with the statement of the Covering Lemma, we provide one more important definition.

**Definition 3** (Covering relation). *Let $f$ be a function and $I_1, I_2$ be two closed intervals. We say that $I_1$ covers $I_2$ under $f$, denoted by $I_1 \xrightarrow{f} I_2$ as long as $I_2 \subseteq f(I_1)$.*

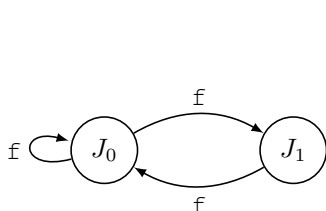
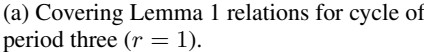
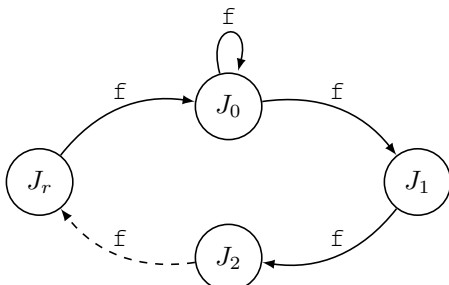

(a) Covering Lemma 1 relations for cycle of period three ($r = 1$).

(b) Covering Lemma 1 relations for cycle of odd period at least three.

Figure 4: The covering relations of intervals $J_0, ..., J_r$ from Lemma 1. Observe that the graph is a directed cycle with a self loop at interval $J_0$. Note that there might be more relations ("edges").

For example, the triangle wave $t(x; 2)$ that has the period 3 point $\frac{2}{9}$ (recall $\frac{2}{9} \rightarrow \frac{4}{9} \rightarrow \frac{8}{9} \rightarrow \frac{2}{9}$) naturally defines two intervals $I_1 = [\frac{2}{9}, \frac{4}{9}]$ and $I_2 = [\frac{4}{9}, \frac{8}{9}]$ with the covering relations: $I_1 \xrightarrow{f} I_2$, $I_2 \xrightarrow{f} I_2$ and $I_2 \xrightarrow{f} I_1$.

**Lemma 1** (Covering Lemma for odd periods). *Let $f : [0, 1] \rightarrow [0, 1]$ be a continuous function and assume $f$ has a cycle $C$ of period $n$, where $n > 1$ is an odd number. Denote $\beta_0, ..., \beta_{n-1} \in C$ the elements of the cycle in increasing order and define the sequence of closed intervals $I_0, ..., I_{n-2}$ where $I_i = [\beta_i, \beta_{i+1}]$ (they have pairwise disjoint interiors). Then, there exists a sub-collection of the aforementioned intervals (not necessarily in the same ordering) $J_0, ...J_r$ with $1 \leq r \leq n - 2$ such that the following covering relation holds:*

1. *$J_i \xrightarrow{f} J_{i+1}$, for $1 \leq i \leq r - 1$,*

2. *$J_r \xrightarrow{f} J_0$ and $J_0 \xrightarrow{f} J_0 \cup J_1$.*

For a pictorial illustration of the Covering Lemma, see Figure 4. In particular, observe that for $n = 3$ we get $r = 1$ so the covering relation is as in Figure 4. We conclude this section with the formal definition of *crossings* (or oscillations) and we refer the reader to Figure 5 for some examples.

**Definition 4** (Crossings). *We say that a continuous function $f : [0, 1] \rightarrow [0, 1]$ crosses the interval $[x, y]$ with $x, y \in [0, 1]$ if there exist $a, b$ such that $f(a) = x$ and $f(b) = y$. Moreover we denote $C_{x,y}(f)$ the number of times $f$ crosses $[x, y]$. That is $C_{x,y}(f) = t$ if there exist numbers $a_1, b_1 < a_2, b_2 < \cdots < a_t, b_t$ in $[0, 1]$ so that $f(a_i) = x$ and $f(b_i) = y$ for all $1 \leq i \leq t$. Observe that if $\mathcal{I}_{f,x,y}$ is used to denote[4] the number of intervals the function $\tilde{f}_{x,y}(z) := \mathbf{1}[f(z) \geq \frac{x+y}{2}]$ is piecewise constant and partitions $[0, 1]$, then $C_{x,y}(f) \leq \mathcal{I}_{f,x,y}$.*

## 3 Periods Determine the Number of Crossings

### 3.1 Period that is not a power of two implies exponential crossings

In this section, we prove our main theorem, the statement of which is given below. Technically, we make use of the Lemma 1 (Covering Lemma) to show the exponential growth of the number of crossings.

**Theorem 3.1.** *Let $f : [0, 1] \rightarrow [0, 1]$ be a continuous function. Assume that there exists a cycle of period $n$ where $n = m \cdot p$, $p$ is an odd number greater than one and $m$ being a power of two (it might be $m = 1$). It holds that there exist $x, y \in [0, 1]$ so that $C_{x,y}(f^{mt})$ is $c^t$ for all $t \in \mathbb{N}^*$, where $c$ is the positive root greater than one of the polynomial equation $\lambda^{p-1} - \lambda^{p-2} - 1 = 0$.*

---

[4]In Telgrasky's paper, $\mathcal{I}_f$ is used to denote the number of intervals where $\mathbf{1}[f(z) \geq \frac{1}{2}]$ is piecewise constant and partitions $[0, 1]$.

**Counting the number of oscillations.** For a given continuous function $f : [0,1] \to [0,1]$, let $J_0, \ldots, J_r$, where $1 \le r \le n - 2$, be the intervals as promised from Lemma 1. We define a sequence of vectors $\delta^t \in \mathbb{N}^{r+1}$ such that $\delta_i^t$ is defined as the number of times the function $f^t$ crosses the interval $J_i$ for all $0 \le i \le r$. In particular we define $f^0$ to be the identity function and hence $\delta^0 = (1, \ldots, 1)$ (all ones vector). For what follows, we will try to express recursively $\delta^t$ in terms of $\delta^{t-1}$ and in the end we will show that $\delta_0^k$ is $\Omega(c^k)$ where $c$ is some constant that depends on $r$. To build some intuition, we first analyze the case of period three and then we prove the general case.

### 3.1.1 WARM UP: THE CASE OF PERIOD 3 AND THE FIBONACCI SEQUENCE

Assume that $f$ has a cycle of period 3, that is the numbers $\{x_0, f(x_0), f^2(x_0)\}$ are distinct and $f^3(x_0) = x_0$ for some $x_0 \in [0,1]$. Let $\beta_0 < \beta_1 < \beta_2$ be the numbers $x_0, f(x_0), f^2(x_0)$ in increasing order. We define $I_0 = [\beta_0, \beta_1]$ and $I_1 = [\beta_1, \beta_2]$. From Lemma 1, when $n = 3$, we can see that $r = 1$ and thus we have the following possibilities for the covering relations:

- Either $I_0 \xrightarrow{f} I_0 \cup I_1$,

- or $I_1 \xrightarrow{f} I_0 \cup I_1$.

We define $J_0$ to be the interval among $I_0, I_1$ that involves the self-loop covering and $J_1$ to be the remaining interval. Define $\delta^t \in \mathbb{N}^2$ as above, and so we get that:

$$\begin{pmatrix} \delta_0^{t+1} \\ \delta_1^{t+1} \end{pmatrix} \ge \begin{pmatrix} 1 & 1 \\ 1 & 0 \end{pmatrix} \begin{pmatrix} \delta_0^t \\ \delta_1^t \end{pmatrix}, \tag{3.1}$$

where $\delta_0^0 = 1$ and $\delta_1^0 = 1$. The matrix $A := \begin{pmatrix} 1 & 1 \\ 1 & 0 \end{pmatrix}$ can be interpreted as the adjacency matrix that corresponds to the covering relations between $J_0, J_1$ (which consists of a directed cycle with a self-loop at vertex $J_0$). The reason we have an inequality instead of an equality is because the Covering Lemma only guarantees that the number of times $J_0$ "covers" $J_0$ and $J_1$ is at least one and not necessarily exactly one.

We set $\alpha^0 = \delta^0$ and we define $\alpha^{t+1} = A\alpha^t$. It is clear that $\delta^t \ge \alpha^t$ (entry-wise) for all $t \in \mathbb{N}$. Moreover, $\alpha_0^t$ is the well-known Fibonacci sequence $F_{t+1}$ (with $F_0 = F_1 = 1$), therefore $\alpha_0^t = \frac{\left(\frac{1+\sqrt{5}}{2}\right)^{t+2} - \left(\frac{1-\sqrt{5}}{2}\right)^{t+2}}{\sqrt{5}}$. We conclude that $\delta_0^t \ge \left(\frac{1+\sqrt{5}}{2}\right)^t$. See also Figure 5 for a pictorial illustration about the proof for $t = 1, 2, 3, 4$.

### 3.1.2 EVERY PERIOD GREATER THAN 3 BUT NOT POWER OF TWO

Assume that $f$ has a cycle of period $n > 3$ with $n$ odd, that is the numbers $\{x_0, f(x_0), f^2(x_0), ..., f^{n-1}(x_0)\}$ are distinct and $f^n(x_0) = x_0$ for some $x_0 \in [0,1]$. Let $\beta_0 < \beta_1 < \beta_2 < ... < \beta_{n-1}$ be the numbers $x_0, f(x_0), f^2(x_0), ..., f^{n-1}(x_0)$ in increasing order. We define $I_i = [\beta_i, \beta_{i+1}]$ for $0 \le i \le n - 2$. From Lemma 1 it follows that there is a subcollection of the intervals $I_0, ..., I_{n-2}$ (with not necessarily the same ordering) $J_0, ..., J_r$ ($1 \le r \le n-2$) such that

1. $J_i \xrightarrow{f} J_{i+1}$, for $1 \le i \le r - 1$,

2. $J_r \xrightarrow{f} J_0$ and $J_0 \xrightarrow{f} J_0 \cup J_1$.

The interval $J_0$ is the one that involves the self-loop covering. As in the case for $n = 3$, we define $\delta^t$ which is in $\mathbb{N}^{r+1}$, with $\delta_i^t$ capturing the number of times $f^t$ crosses the interval $J_i$. We get that:

$$\begin{pmatrix} \delta_0^{t+1} \\ \delta_1^{t+1} \\ \vdots \\ \delta_r^{t+1} \end{pmatrix} \ge A \begin{pmatrix} \delta_0^t \\ \delta_1^t \\ \vdots \\ \delta_r^t \end{pmatrix}, \tag{3.2}$$

where $\delta^0 = (1, \ldots, 1)$ (all ones vector) and $A \in \mathbb{R}^{(r+1)\times(r+1)}$ is defined to be:

$$
\begin{cases}
A_{ji} = 1, & \text{if } i = 0, j = 0 \\
A_{ji} = 1, & \text{if } j = i+1 \text{ and } 0 \leq i \leq r-1 \\
A_{ji} = 1, & \text{if } i = r, j = 0 \\
A_{ji} = 0, & \text{otherwise}
\end{cases}
\tag{3.3}
$$

In words, $A$ is the adjacency matrix of a graph with $r+1$ nodes that is a directed cycle that involves a self-loop at vertex $J_0$. We define $\alpha^t$ in a similar way as in the case for period three, i.e., $\alpha^{t+1} = A\alpha^t$ and $\alpha^0 = \delta^0$ so that $\delta^t \geq \alpha^t$ (entry-wise) for all $t \in \mathbb{N}$. We can easily observe that the following holds: $\alpha^{t+1} = A^{t+1}\alpha^0$.

Our next plan is to compute a lower bound on the spectral radius of the matrix $A^\top$ (denoted by $\mathrm{sp}(A^\top)$) with the following claim (proof in Appendix A).

**Claim 3.1.** *The characteristic polynomial of $A^\top$ is:*

$$
\pi(\lambda) = \lambda^{r+1} - \lambda^r - 1. \tag{3.4}
$$

Let us call $\rho_r$ the largest root in absolute value of the polynomial $\pi(\lambda)$ in A.1. Since $A$ is a non-negative matrix, the largest root in absolute value is actually a positive real number (by the Perron-Frobenius theorem). It is easy to see that the polynomial in A.1 has always a root greater than one and less than two (by Bolzano's theorem, see $\pi(1) = -1 < 0$ and $\pi(2) = 2^{r+1} - 2^r - 1 = 2^r - 1 > 0$).

Hence we have $\mathrm{sp}(A) = \rho_r > 1$. Furthermore, it is easy to see that since $A$ is a non-negative matrix (and powers of $A$ are also non-negative), it holds that

$$
\left\| A^t \right\|_\infty = \sum_{j=0}^r A_{0j}^t
$$

for all $t \geq 1$, that is the row with the largest sum of its entries is the first row (row for $i = 0$). Using the fact that

$$
\left\| A^t \right\|_\infty \geq \mathrm{sp}(A^t) = \rho_r^t,
$$

that is the spectral radius of a matrix is always at most any matrix norm, we conclude that $\sum_{j=0}^r A_{0j}^t \geq \rho_r^t$.

The case of odd period greater than three follows by noting that $\sum_{j=0}^r A_{0j}^t = \alpha_0^t$, thus $\delta_0^t \geq \alpha_0^t \geq \rho_r^t$. Observe that for period three, we have that $r = 1$ and also $\rho_1 = \frac{1+\sqrt{5}}{2}$ (the largest root of $\lambda^2 - \lambda - 1 = 0$).

We would like to make the following two remarks:

*Remark* 3.1. The spectral radius $\rho_r$ is strictly decreasing in $r$: this is easy to see since $\rho_r > 1$ and is satisfying the equation $x^{r+1} - x^r = 1$ (note that $x^{r+1} - x^r$ is increasing in $r$ for $x > 1$). This implies that smaller odd periods can potentially have a number of crossings that grows at faster rates than larger odd periods, hence giving rise to more complex behaviors. See also Remark 4.1.

*Remark* 3.2 (The case of even period but not power of two). Our result above is applied for cycles of period $n = m \cdot n'$ where $m$ is a power of two and $n'$ is an odd number greater than one. The trick is to observe that if a function has cycle of period $n$, then $f^m$ has a cycle of period $n'$ (which is an odd number greater than one). Therefore, the number of oscillations $C_{x,y}(f^{mt})$ with $x, y$ being the endpoints of $J_0$, is at least $\rho_{n'-2}^t$ for $t \in \mathbb{N}$.

*Proof of Theorem 3.1.* The proof now follows from the case analysis carried out in Sections 3.1.1, 3.1.2 and Remark 3.2. $\qquad\square$

## 3.2 PERIOD THAT IS A POWER OF TWO MAY HAVE POLYNOMIAL CROSSINGS

**Lemma 2** (Period power of two - proof in Appendix A). *There exist continuous functions $f$ with prime period $n$ that is a power of two so that the number of crossings $C_{x,y}(f^t)$ scales at most polynomially with $t$ for any $x, y \in [0, 1]$.*

## 4 PERIOD-DEPENDENT LOWER BOUNDS FOR DNNS

Building on Telgarsky (2015; 2016), the representation power of different networks will be measured via the classification error. For a given collection of $n$ points $(x_i, y_i)_{i=1}^n$ with $y_i \in \{0, 1\}$, one can define the classification error of a function $g$ to be ($\tilde{g}$ is just the thresholded value of $g$):

$$\mathcal{R}(g) = \frac{1}{n} \sum_{i=1}^n \mathbf{1}[\tilde{g}(x_i) \neq y_i]$$

In this section, we argue that functions with cycles of period not a power of two, will have compositions for which any shallow neural network will have classification error a positive constant.

Assume we are given a continuous function $f : [0, 1] \to [0, 1]$ so that $f$ has a cycle of period $m \times p$ where $p$ is an odd number greater than one and $m$ is a power of two. From Theorem 3.1, there exist $x, y \in [0, 1]$ so that $\mathrm{C}_{x,y}(f^{tm})$ is at least $\frac{\rho_{p-2}^t}{2}$, where $\rho_r$ is defined to be the root that is greater than one of the polynomial equation $\lambda^{r+1} - \lambda^r - 1 = 0$. We set $\rho := \rho_{p-2}$, $h := f^{k \cdot m}$ and assume that $g : [0, 1] \to [0, 1]$ is a neural network with $l$ layers and $u$ nodes (ReLU activations) per layer. In Lemma 2.1 of Telgarsky (2015), it is proved that a neural network with $u$ ReLU units per layer and with $l$ layers is piecewise affine with at most $(2m)^l$ pieces.

We define as $\tilde{h}(z) = \mathbf{1}[h(z) \geq \frac{x+y}{2}]$ and $\tilde{g}(z) = \mathbf{1}[g(z) \geq \frac{x+y}{2}]$ (note that we changed the threshold to be $\frac{x+y}{2}$ instead of $\frac{1}{2}$ that was used in Telgarsky (2015)).

Since $\mathrm{C}_{x,y}(h)$ is at least $\rho^k$, it holds that there exist points $(x_i, y_i)_{i=1}^{2n}$ with $n := \frac{\lfloor \rho^k \rfloor}{2}$ such that $h(x_j) = x$, $y_j = 0$ for $j$ odd and $h(x_j) = y$, $y_j = 1$ for $j$ even. It is clear that for this collection of points the classification error of the function $h$ is zero, whereas the classification error for function $g$ is bounded from below by

$$\mathcal{R}(g) \geq \frac{n - 4(2u)^l}{2n} = \frac{1}{2} - \frac{(2u)^l}{n}.$$

The above inequality is an application of Lemma 2.2 of Telgarsky (2015) (with careful counting it has been slightly improved). By choosing $u$ to be at most $\frac{\rho^{\frac{k}{l}}}{8}$ it holds that the classification error $\mathcal{R}(g) \geq \frac{1}{4}$ for any neural network $g$ with $u$ ReLUs and $l$ layers.

The above discussion implies the following theorem:

**Theorem 4.1** (Classification Error Theorem). *Let $k$ be a positive integer and $f$ be a function of period $m \times p$ with $p$ an odd number greater than one and $m$ being a power of two (it might hold $m = 1$). We set $\rho$ to be the positive root greater than one of the polynomial equation $\lambda^{p-1} - \lambda^{p-2} - 1 = 0$. We can construct a sequence of points $(x_i, y_i)_{i=1}^{2n}$ with $n := \frac{\lfloor \rho^k \rfloor}{2}$ so that the classification error of function $f^{mk}$ is zero, whereas the classification error of any neural network of $l$ layers and $u$ nodes per layer with $u \leq \frac{\rho^{\frac{k}{l}}}{8}$ satisfies $\mathcal{R}(g) \geq \frac{1}{4}$.*

*Remark* 4.1. Observe that if the number of units $u$ per layer is constant and the number of layers $l$ is $o(k)$, then the classification error is always a positive constant for any neural network (whereas for $f^{mk}$ is zero). Moreover, observe that since $\rho$ is decreasing in $p$ (recall $p$ is the odd factor of the period), it holds that the classification error decreases as $p$ increases (with fixed number of layers and nodes per layer). This indicates that the composition of functions with large odd period is simpler than of functions with small odd period (period greater than one) following the intuition we have from the Sharkovsky's ordering.

### ACKNOWLEDGMENTS

Vaggos Chatziafratis is partially supported by an Onassis Foundation Scholarship. Sai Ganesh Nagarajan would like to acknowledge SUTD President's Graduate Fellowship (SUTD-PGF). Ioannis Panageas and Xiao Wang would like to acknowledge NRF-NRFFAI1-2019-0003, SRG ISTD 2018 136 and ANR NRF 0095 ALIAS. Part of this project happened while the authors were visiting the Simons Foundation program "Foundations of Deep Learning" at Berkeley and would like to thank the organizers for their hospitality.

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

## A  APPENDIX

**Claim A.1.** *The characteristic polynomial of $A^\top$ is:*

$$\pi(\lambda) = \lambda^{r+1} - \lambda^r - 1. \tag{A.1}$$

*Proof.* Let $I$ denote the identity matrix of size $(r+1) \times (r+1)$. We consider the matrix:

$$A^\top - \lambda I = \begin{pmatrix} 1-\lambda & 1 & 0 & 0 & 0 & \ldots & 0 \\ 0 & -\lambda & 1 & 0 & 0 & \ldots & 0 \\ 0 & 0 & -\lambda & 1 & 0 & \ldots & 0 \\ \vdots & \vdots & \vdots & \vdots & \vdots & \vdots & \vdots \\ 0 & 0 & 0 & \ldots & 0 & -\lambda & 1 \\ 1 & 0 & 0 & 0 & \ldots & 0 & -\lambda \end{pmatrix}.$$

Observe that $\lambda = 0, 1$ are not eigenvalues of the matrix $A^\top$., hence we can multiply the first row by $\frac{1}{\lambda-1}$, the second row by $\frac{1}{\lambda(\lambda-1)}$, the third row by $\frac{1}{\lambda^2(\lambda-1)}$,..., the $i$-th row by $\frac{1}{\lambda^{i-1}(\lambda-1)}$ (and so on) and add them to the last row. Let $B$ be the resulting matrix:

$$B = \begin{pmatrix} 1-\lambda & 1 & 0 & 0 & 0 & \ldots & 0 \\ 0 & -\lambda & 1 & 0 & 0 & \ldots & 0 \\ 0 & 0 & -\lambda & 1 & 0 & \ldots & 0 \\ \vdots & \vdots & \vdots & \vdots & \vdots & \vdots & \vdots \\ 0 & 0 & 0 & \ldots & 0 & -\lambda & 1 \\ 0 & 0 & 0 & 0 & \ldots & 0 & -\lambda + \frac{1}{\lambda^{r-1}(\lambda-1)} \end{pmatrix}.$$

It is clear that $\det(B) = 0$ as an equation has the same roots as $\det(A^\top - \lambda I) = 0$. Since $B$ is an upper triangular matrix, it follows that

$$\det(B) = (-\lambda)^{r-1}(1-\lambda)\left(-\lambda + \frac{1}{\lambda^{r-1}(\lambda-1)}\right).$$

We conclude that the eigenvalues of $A^\top$ (and hence of $A$) must be roots of $(\lambda^r - \lambda^{r-1})\lambda - 1$ and the claim follows. □

**Lemma 3** (Period power of two). *There exist continuous functions $f$ with prime period $n$ that is a power of two so that the number of crossings $C_{x,y}(f^t)$ scales at most polynomially with $t$ for any $x, y \in [0,1]$.*

*Proof.* The easiest example one can construct is the function $f : [0,1] \to [0,1]$ that is defined $f(x) = 1 - x$. Observe that for any $a \in [0,1]$ one has $f(f(a)) = a$ and moreover if $a \neq \frac{1}{2}$ then $f(a) \neq a$. Hence $f$ is a function of prime period two. It is also clear that $f^t(x) = x$ if $t$ is even and $f^t(x) = 1 - x$ if $t$ is odd, so the number of crossings is always one for all $t \in \mathbb{N}^*$.

One other less trivial example is the following function (see also Figure 6):

$$f(x) = \begin{cases} -x + 5, & 1 \leq x \leq 2 \\ -2x + 7, & 2 \leq x \leq 3 \\ x - 2, & 3 \leq x \leq 4. \end{cases}$$

It is not hard to see that this function has prime period four ($f(1) = 4$, $f(4) = 2$, $f(2) = 3$, $f(3) = 1$). Let $J_0 = [1,2]$, $J_1 = [2,3]$, $J_2 = [3,4]$. It is clear that

- $f(J_0) = J_2$, $f(J_1) = J_0 \cup J_1$ and $f(J_2) = J_0$.

By letting $\delta_i^t$ be the number of crossings of the function $f$ for the interval $J_i$ ($i \in \{0,1,2\}$), one has recursively

$$\begin{pmatrix} \delta_0^{t+1} \\ \delta_1^{t+1} \\ \delta_2^{t+1} \end{pmatrix} = \begin{pmatrix} 0 & 1 & 1 \\ 0 & 1 & 0 \\ 1 & 0 & 0 \end{pmatrix} \begin{pmatrix} \delta_0^t \\ \delta_1^t \\ \delta_2^t \end{pmatrix} \tag{A.2}$$

where $\delta^0 = (1,1,1)$ (all ones vector). It is easy to observe that the matrix $A = \begin{pmatrix} 0 & 1 & 1 \\ 0 & 1 & 0 \\ 1 & 0 & 0 \end{pmatrix}$ has spectral radius one (as opposed to the case of odd period greater than one) and moreover it holds

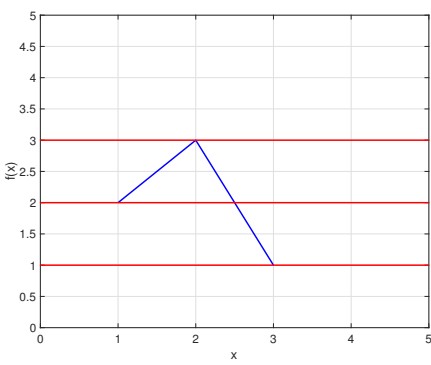

(a) This figure captures one composition of function $f$. Observe that $f$ crosses the interval $[2, 3]$ two times (once for $x \in [1, 2]$ and once for $x \in [2, 3]$) and it crosses the interval $[1, 2]$ once. In particular, $\delta^1 = (2, 1)$.

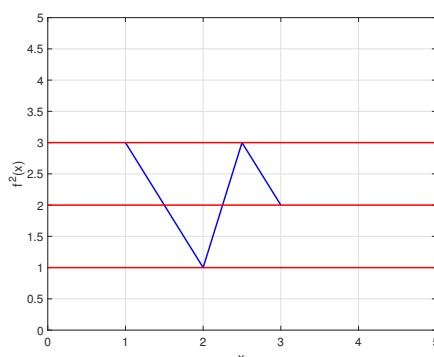

(b) This figure captures two compositions of function $f$. Observe that $f$ crosses the interval $[2, 3]$ three times (two times for $x \in [2, 3]$ and once for $x \in [1, 2]$) and it crosses the interval $[1, 2]$ two times (once for $x \in [1, 2]$ and once for $x \in [2, 3]$). In particular, $\delta^2 = (3, 2)$.

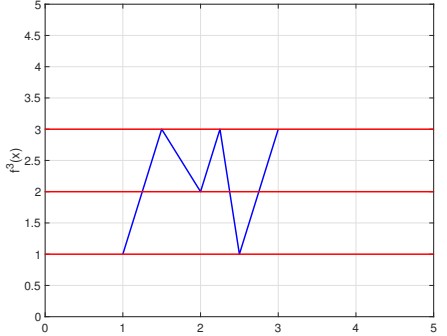

(c) This figure captures three compositions of function $f$. Observe that $f$ crosses the interval $[2, 3]$ five times (three times for $x \in [2, 3]$ and twice for $x \in [1, 2]$) and it crosses the interval $[1, 2]$ three times (once for $x \in [1, 2]$ and twice for $x \in [2, 3]$). In particular, $\delta^3 = (5, 3)$.

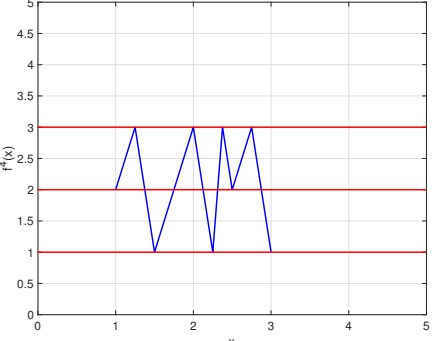

(d) This figure captures four compositions of function $f$. Observe that $f$ crosses the interval $[2, 3]$ eight times (five times for $x \in [2, 3]$ and three times for $x \in [1, 2]$) and it crosses the interval $[1, 2]$ five times (twice for $x \in [1, 2]$ and three times for $x \in [2, 3]$). In particular, $\delta^4 = (8, 5)$.

Figure 5: Compositions of a piecewise linear function that has a point of period 3.

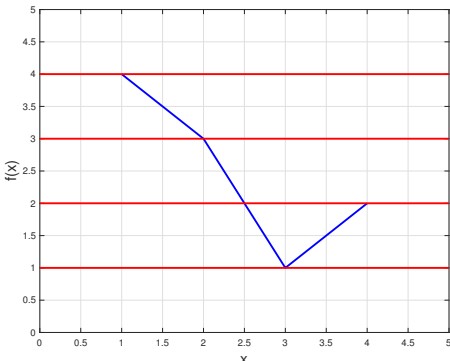

Figure 6: A piecewise linear function $f : [1, 4] \to [1, 4]$ that has prime period four.

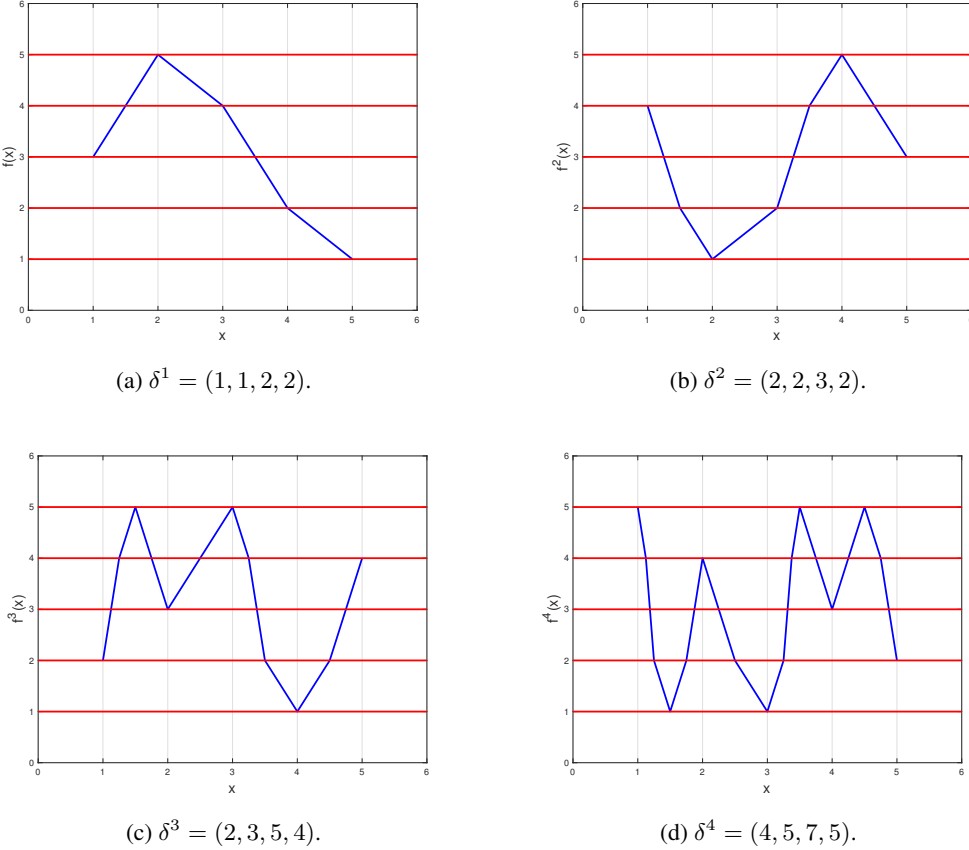

(a) $\delta^1 = (1, 1, 2, 2)$.

(b) $\delta^2 = (2, 2, 3, 2)$.

(c) $\delta^3 = (2, 3, 5, 4)$.

(d) $\delta^4 = (4, 5, 7, 5)$.

Figure 7: Compositions of a piecewise linear function that has a point of period 5. We start with the all ones vector for $\delta^0$ and each composition arises from the covering relation between the sets.

that $\sum_{i=0}^{2} \sum_{j=0}^{2} A_{ij}^t = t + 3$ for all $t \in \mathbb{N}^*$. We conclude that $\alpha_0^t + \alpha_1^t + \alpha_2^t = t + 3$, therefore the number of crossings for $J_0, J_1, J_2$ of the function $f^t$ grows linearly with $t$ (and not exponentially). Since the function we defined is of prime period four and is piecewise monotone (and so is any composition with itself) in each interval $J_0, J_1, J_2$, we conclude that the number of crossings of $f^t$ for any possible pairs of values is at most linear in $t$. □

# B  AN EXAMPLE WHICH IS PERIOD 5 BUT NOT PERIOD 3 (LI & YORKE (1975))

Here, we show an example function, that has a point of period 5, but not period 3, thereby respecting the Sharkovsky ordering. Our proof approach for general odd periods is similar to the case of period 3, by using the induced covering graph and counting the crossings over each interval. This is illustrated in Figure 7.

# C  THE HETEROGENEITY OF THE LOGISTIC MAP

In this section, we illustrate how the compositions of the logistic map $f(x; r) := rx(1 - x)$ behaves as $r$ varies slightly. We give certain examples in the form of Figure 8. It is known that the map when $r = 3.9$, has a point of period 3. In contrast when $r$ is reduced to $3.5$ the map has a point of period 4 and further bringing $r$ down to $3.2$ will ensure that the map has a point of period 2. The figures below illustrate how the oscillations grow under these scenarios.

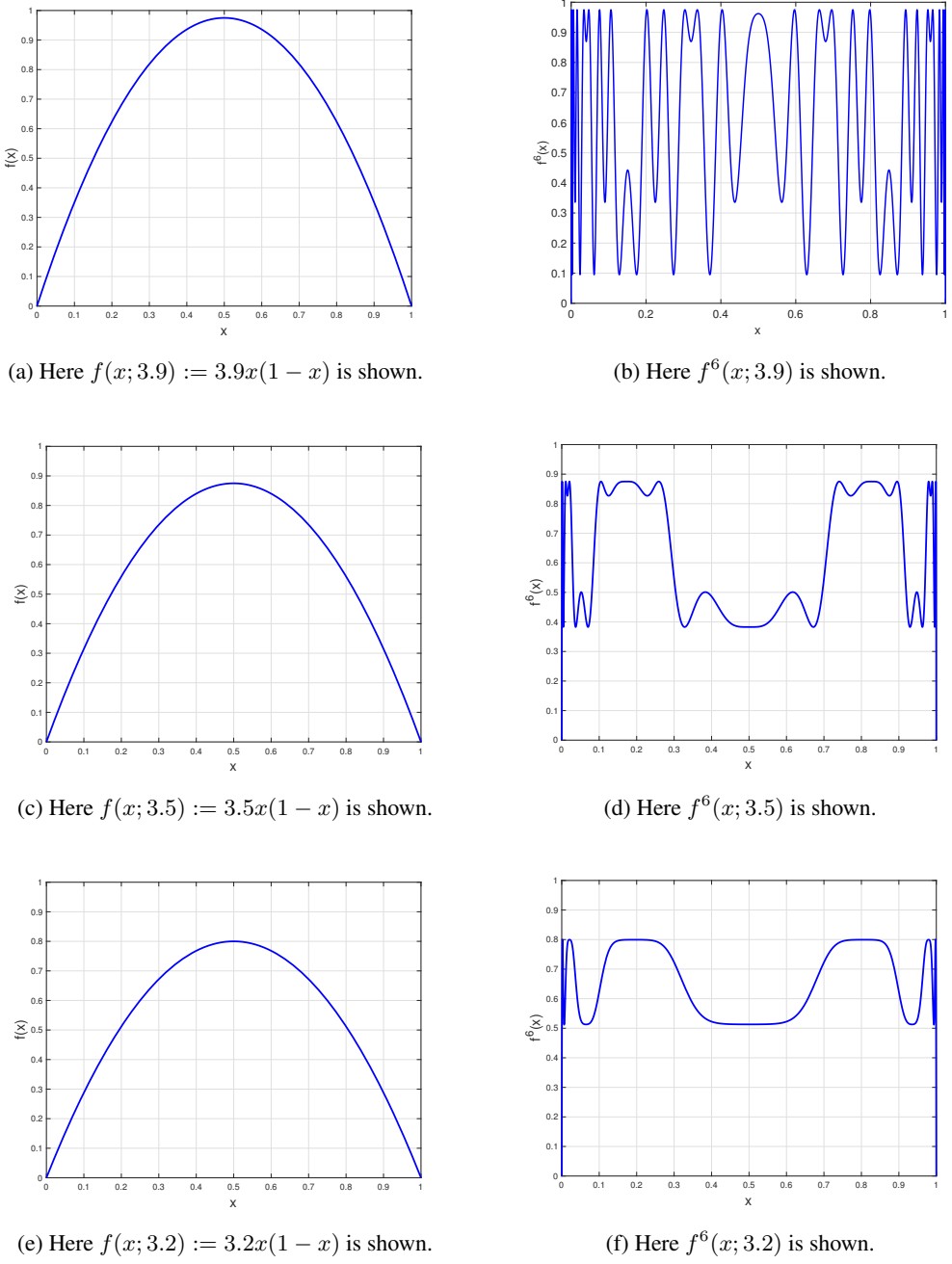

(a) Here $f(x; 3.9) := 3.9x(1 - x)$ is shown.

(b) Here $f^6(x; 3.9)$ is shown.

(c) Here $f(x; 3.5) := 3.5x(1 - x)$ is shown.

(d) Here $f^6(x; 3.5)$ is shown.

(e) Here $f(x; 3.2) := 3.2x(1 - x)$ is shown.

(f) Here $f^6(x; 3.2)$ is shown.

Figure 8: The compositions of the logistic map $f(x; r) := rx(1 - x)$ with different parameter values are shown here. The left column has the functions themselves while the right column shows the corresponding compositions. We can see that oscillations in these family of functions vary vastly with changes in $r$ and these changes are made in the weights of an appropriate neural network (see Telgarsky (2016),Schmitt (2000)).

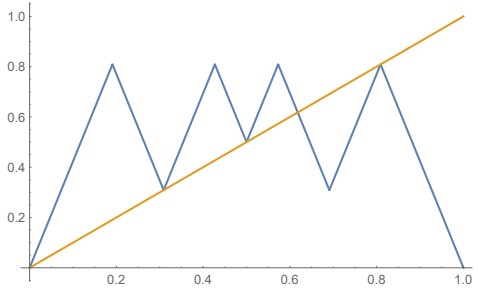
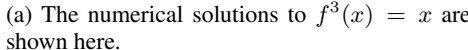

(a) The numerical solutions to $f^3(x) = x$ are shown here.

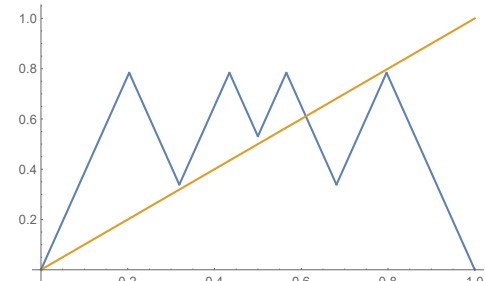

(b) The numerical solutions to $g^3(x) = x$ are shown here.

Figure 9: We see that $f^3(x) = x$ has solutions other than just the fixed point, since it has intersections in 3 other places, other than the fixed point. However, $g^3(x) = x$ does not have any solutions other than the fixed point, as there are no other intersections.

## D  FURTHER DISCUSSIONS

In this section, we provide some additional theoretical and experimental remarks on our characterization.

### D.1  INCORPORATING BIAS TERMS

If we add a bias term in the ReLU activation unit, e.g., use $\max(v, \epsilon)$ instead of $\max(v, 0)$ for the activation gates, where $\epsilon$ is a small number (positive or negative), then our results do not change; in particular our trade-off in Theorem 4.1 still holds (since the Lemma 2.2 from Telgarsky (2015) is for general sawtooth functions). But, if one adds the bias term to the function $f$ itself, then things get more interesting indeed: Suppose $f$ has some period $p$ where $p$ is not a power of two; due to bifurcation phenomena (i.e., phenomena arising because we are at critical regimes of parameters such as the parameter $\mu$ in our generalized triangle wave function), then the compositions of the function ($f$+bias term) with itself may give rise to qualitatively different behaviors compared to $f$. In particular, the function ($f$+bias term) might not have period $p$ anymore. Intuitively, one can think that the small bias term is amplified after many compositions and is not negligible anymore.

One such example is the triangle function $f(x) = \phi x$ for $0 \le x \le 0.5$ and $\phi(1-x)$ for $1/2 \le x \le 1$, where $\phi = (1 + \sqrt{5})/2$ is the golden ratio. This function has period 3, see Figure 9a. However, if we consider the function $g(x) = (\phi - \epsilon)x$ for $0 \le x \le 0.5$ and $(\phi - \epsilon)(1 - x)$ for $0.5 \le x \le 1$ with $\epsilon > 0$ (arbitrarily small positive) then $g$ does not have period 3, see Figure 9b. In this sense, period as a property can be brittle to numerical changes if we are at the critical point.

### D.2  SOME EXPERIMENTAL EVIDENCE

In this section, we provide experimental evidence for our depth separation results by training a neural network of constant width, but with increasing depth on a classification task that closely resembles the $n$-alternating points problem that appeared in Telgarsky (2015) and is the foundation of our separation results as well. As mentioned before, this is a specific instance of a function that has a point of period 3. For simplicity, we do not consider this original problem exactly but rather a "smoothed" variant of it, in order to make it more amenable to the training procedure. Our goal is to create a diagram showing how the classification error drops as a function of the depth of the network for a fixed value of the width.

We create 8000 equally spaced points from [0,1] (in increasing order), where the first 1000 points are of label 0, the second 1000 are label 1 and this label alternates every 1000 points. This is what we call a "smoothed" alternating point problem. Although, the theory would have used the classical 8-alternating points to argue about the lower bounds, in practice, performing training of deep (4 and above layers) and narrow networks (hidden layers with less than 4 neurons) with very few data points is a major challenge, see for instance Lu et al. (2018). Apart from the separation results that we show in theory, we show empirically that deep networks generally do improve the accuracy in

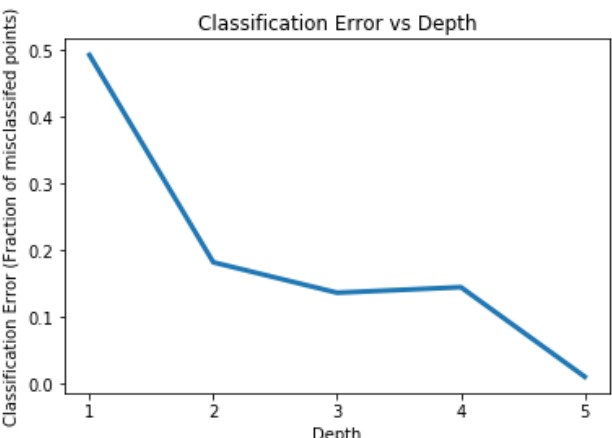

Figure 10: We see that depth does reduce the classification error for this particular task and when depth is 5, the classification error is close to 0. The saturation in between may be attributed to the general uncertainties in the training/optimization.

this task compared to the shallow network and in fact a deep network with 5 layers can reach an accuracy of 99.04%. Any additional uncertainties in the error is generally attributed to the training procedure.

To perform the experiments, we vary the depth of the neural network (excluding the input and the output layer) as $d = 1, 2, 3, 4, 5$. In addition, we fix the neurons for each layer to be 6. All activations are ReLU's, while the last layer is the classifier that uses a sigmoid to output probabilities. Each model adds one extra hidden layer and we make use of the same hyper-parameters to train all networks. Moreover, we require the training error or the classification error to tend to 0 during the training procedure, i.e, we will try and overfit the data (as we try to demonstrate a representation result, rather than a statistical/generalization result). Thus, for the actual training we use the same parameters to train all the different models using the "ADAM" optimizer Kingma & Ba (2014) and make the epochs to be 200 in order to enable overfitting. To record the training error, we verify that the training saturates by seeing the performance over the epochs and report by default the error in the last epoch. The results are shown in Figure 10.

### D.3 PERIOD AS A NATURAL CHARACTERIZATION

In a nutshell, our paper provides a "natural" property of a function (periodic points of certain periods) and then derive depth-width trade-offs based on it. This addresses some questions raised not only in Telgarsky (2016; 2015)'s works, but also in the paper Poole et al. (2016) that seeks to provide a natural, general measure of functional *complexity* helping us understand the benefits of depth. On the contrary, many of the previous depth separation results take a worst case approach for the representation question (showing that there exist functions implemented by deep networks that are hard to approximate with a shallow net). However, it is not clear whether such analysis applies to the typical instances arising in practice of neural-networks. We believe that our work together with Telgarsky (2016; 2015) and the paper Eldan & Shamir (2016) show a depth separation argument for very natural functions, such as the triangle waves or the indicator function of the unit ball.

Given a specific prediction task in practice, how could one assess the period? We believe that this would be extremely useful yet a very difficult question that seems to be outside the reach of current techniques in the literature. Previous works and our work so far are able to present depth separation for representing certain functions.

We point out that, intuitively, our characterization result consists of a **certificate** informing us qualitatively and quantitatively about which functions have complicated compositions and which not. Similar to computational problems in class NP, if one is given the **certificate** (the points $(x_1, \ldots, x_p)$), then one can easily verify (if we have oracle access to evaluate the function $f$), if the given function has a $p$-periodic cycle with points $(x_1, \ldots, x_p)$. Nevertheless, we believe that

finding the **certificate** for arbitrary continuous functions is not a straightforward problem, except maybe for particular restricted classes of functions. Having said that, we want to emphasize that in many prediction problems that are inspired by physics, one may a priori expect to have complicated dynamics behavior and hence require deeper networks for better performance. Such examples include efforts to solve the notorious 3-body problem or turbulent flows showing empirical evidence that complex physical processes require deep networks (see for instance, Ling et al. (2016) and Breen et al. (2019) that uses a 10 layered neural network).

