# OpenReview forum: "Depth-Width Trade-offs for ReLU Networks via Sharkovsky's Theorem"
_ICLR.cc/2020/Conference — Accept (Spotlight)_

### Official Review · AnonReviewer2 · 2019-10-25
**Official Blind Review #2**

**Rating:** 8

**Review:**

The paper studies how the expressive power of NN depends on its depth and width. Sharkovsky's theorem is leveraged to characterize the depth-width tradeoff in the ability of ReLU networks to represent functions with periodic points. A lower bound on the depth necessary to represent periodic functions is also provided. All in all, the paper furthers the understanding on the benefit of deep nets for representing certain function classes.

I found this to be a serious and well-written paper. The application of Sharkovsky's results is clever and well in place. My main criticism has to do with the structure, which I think overloads with general theory before getting to the main point the paper is making. I suggest stating Theorem 4.1 earlier, even as soon as Section 1.3, and use the discussion therein as an interpretation of the result. All the technical details, such as definition, Sharkovsky's Thm and proofs, can follows after than. The theoretical background is very interesting, but it would be better to start from the contribution to ML and get into the math later on.

The period dependent depth lower bound is nice but not very useful. Given a certain classification task, how could one assess/bound/approximate the period? This is general issue with this type of theory -- while it broadens our understanding it is hard to put it into actual use.

Another small comment: it would be useful to provide intuition for some of the definitions in the paper. For example Def. 3 lacks such.

**Experience Assessment:**

I have read many papers in this area.

**Review Assessment: Checking Correctness Of Derivations And Theory:**

I assessed the sensibility of the derivations and theory.

**Review Assessment: Checking Correctness Of Experiments:**

I assessed the sensibility of the experiments.

**Review Assessment: Thoroughness In Paper Reading:**

I read the paper at least twice and used my best judgement in assessing the paper.

---

> ### Author Response · Authors · 2019-11-15
> **Response to Reviewer #2**
>
> First we thank Reviewer 2 for their time, positive feedback and valuable comments.
>
> Following the reviewer’s suggestion, we have restructured the paper in the newer version in such a way so that the main results and the contribution to ML come first, before the technical details based on dynamical systems theory. Of course, feel free to suggest any other change you think would improve the current write-up. We also provided an example for Definition 2 and Definition 3 to make the presentation cleaner. We also added a discussion section in the Appendix with several additions/comments.
>
> Regarding the question of usefulness:
>
> In terms of theoretical advantages, our paper in a nutshell gives a *natural* property of a function (periodic points of certain periods) and then derives depth-width trade-offs based on it. This addresses some questions raised in Telgarsky’s work, but also in the paper “Exponential expressivity in deep neural networks through transient chaos” (https://papers.nips.cc/paper/6322-exponential-expressivity-in-deep-neural-networks-through-transient-chaos) that seeks to provide a natural, general measure of functional complexity helping us understand the benefits of depth. On the contrary, many of previous depth separation results take a worst case approach for the representation question (showing that there exist functions implemented by deep networks that are hard to approximate with a shallow net). However, it is not clear whether such analysis applies to the typical instances arising in practice of neural-networks. We believe that our work together with Telgasky’s and the paper “The power of depth for feedforward neural networks” (by Eldan/Shamir) show a depth separation argument for very natural functions, like the triangle waves or the indicator function of the unit ball.
>
> Continuing with the question of given a specific prediction task, how could one assess the period, we agree that this would be extremely useful in practice but this is indeed a very difficult question that seems to be outside the reach of current techniques in the literature. Previous works and our work so far are able to present depth separation for representing certain functions.
>
> We would like to point out that, intuitively, our characterization result consists of a certificate informing us qualitatively and quantitatively about which functions have complicated compositions and which not. Similar to computational problems in class NP, if one is given the certificate (the points x_1,...,x_p), then one can easily verify if the given function has a p-periodic cycle with points x_1,...,x_p (given oracle access to the function). Nevertheless, we believe that finding the certificate for arbitrary continuous functions is not a straightforward problem except maybe for restricted classes of functions. Having said that, we want to emphasize that in many prediction problems that are inspired by physics, one may a priori expect to have complicated dynamics behaviour and hence requiring deeper networks for better performance. Such examples include efforts to solve the notorious 3-body problem or turbulent flows showing empirical evidence that complex physical processes require deep networks (see for instance, “Reynolds averaged turbulence modelling using deep neural networks with embedded invariance” (2016) and “Newton vs the machine: solving the chaotic three-body problem using deep neural networks” (2019) (https://arxiv.org/pdf/1910.07291.pdf) where they use a 10 layered neural network.)

---

### Official Review · AnonReviewer1 · 2019-10-25
**Official Blind Review #1**

**Rating:** 8

**Review:**

In tackling a curious construction by Telgarsky regarding a certain class of functions that can be represented by deep networks (but not shallow networks (unless those shallow networks have exponentially many units)), the authors derive depth-width tradeoff conditions for when relu networks are able to represent periodic functions using dynamical systems analysis.

This paper was a delight to read.  I particularly enjoyed the motivating examples, and the clean exposition of Sharkovsky's theorem.  This result seems to cleanly answer the open question originally posed by Telgarsky, and the proofs are cleanly written, and correct to my (admittedly not perfect) knowledge.  I strongly suggest acceptance.

Questions/comments:

1. Could the author speculate on how the introduction of a bias term might affect their lower bound?  Presumably, this breaks the cleanness of the characteristic polynomial for $A$, but perhaps there are limits where it's still tractable?  This analysis certainly isn't necessary for publishing--I'm simply curious.

2. Could the authors provide some guiding intuition for the sharpness of their lower bound? (possibly on a synthetic dataset?) . I'm particularly imagining a plot that literally shows "classification error" versus "depth" for some fixed task.  While this is certainly a strong theoretical result, it would be nice to be able to contextualize how this result actually shines for a "real" model (and would help me believe the result "in my gut" so to speak).

**Experience Assessment:**

I do not know much about this area.

**Review Assessment: Checking Correctness Of Derivations And Theory:**

I assessed the sensibility of the derivations and theory.

**Review Assessment: Checking Correctness Of Experiments:**

I assessed the sensibility of the experiments.

**Review Assessment: Thoroughness In Paper Reading:**

I read the paper at least twice and used my best judgement in assessing the paper.

---

> ### Author Response · Authors · 2019-11-15
> **Response to Reviewer #1**
>
> First we thank Reviewer 1 for their time, positive feedback and valuable comments. Both questions the reviewer asked are very interesting and important and we answer them below.
>
> Addressing the first question regarding the bias term:
>
> If the reviewer asks about adding a bias term in the ReLU activation unit, e.g., use \max(v,\epsilon) instead of  \max(v,0) for the activation gates, where \epsilon is a small number (positive or negative), then our results do not change; in particular our tradeoff in Theorem 4.1 still holds (since the Lemma 2.2 from Telgarsky is for general sawtooth functions). If the reviewer asks about what happens if one adds the bias term to the function f itself, then things get more interesting indeed: Suppose f has some period p where p is not a power of two; due to bifurcation phenomena (i.e., phenomena arising because we are at critical regimes of parameters like the \mu parameter in our generalized triangle wave function), then the compositions of the function (f+bias term) with itself may give rise to different behaviours qualitatively compared to f. In particular, the function (f+bias term) might not have period p anymore. Intuitively you can think that the small bias term is amplified after many compositions and is not negligible anymore.
>
> Such a brittle example is the triangle function f(x) = \phi * x for 0<=x<=½ and \phi(1-x) for 1/2<=x<=1 where \phi = (1+\sqrt{5})/2 is the golden ratio. This function is easy to see that has period 3 (we include illustrative figures in the newer version of the paper). However, if we consider the function g(x) = (\phi-\epsilon)x for 0<=x<=½ and (\phi-\epsilon)(1-x) for 1/2<=x<=1 with \epsilon>0 (arbitrarily small positive) then g does not have period 3. In this sense, period as a property can be brittle to numerical changes if we are at the critical point. We updated the newer version as well with this brittle example.
>
> Addressing the second question regarding empirical intuition and performance for the classification error:
>
> Here we performed experiments on the synthetic dataset generated by the triangle functions and trained neural networks with different depths (both at regimes where representation is possible and at regimes where it is impossible). We plotted the classification error as a function of the depth as the reviewer suggested and we include the figures in the newer version of the paper. Some details for the experimental setup are included as well. The code together with the final figure are added to the google doc containing our code for the ICLR submission (we added a Python Script for the Neural Network experiment (To be run in ipython 3 enivronment)).

---

> > ### Comment · AnonReviewer1 · 2019-11-15
> > **Response to author's comment**
> >
> > I greatly appreciate the author's thorough response!  I also appreciate the inclusion of the synthetic dataset!  Unfortunately, it isn't possible for me to raise my score any higher (because it is already at the maximum).

---

### Decision · Program_Chairs · 2019-12-19

**Decision:**

Accept (Spotlight)

**Comment:**

The article is concerned with depth width tradeoffs in the representation of functions with neural networks. The article presents connections between expressivity of neural networks and dynamical systems, and obtains lower bounds on the width to represent periodic functions as a function of the depth. These are relevant advances and new perspectives for the theoretical study of neural networks. The reviewers were very positive about this article. The authors' responses also addressed comments from the initial reviews.